# Drug Repurposing for Influenza Virus Polymerase Acidic (PA) Endonuclease Inhibitor

**DOI:** 10.3390/molecules26237326

**Published:** 2021-12-02

**Authors:** Xin Meng, Ye Wang

**Affiliations:** 1School of Life Science, Jilin University, No. 2699 Qianjin Street, Changchun 130012, China; 18624395651@163.com; 2School of Life Science, Jilin Normal University, Siping 136000, China

**Keywords:** lifitegrast, virtual screening, drug repurposing

## Abstract

Drug repurposing can quickly and effectively identify novel drug repurposing opportunities. The PA endonuclease catalytic site has recently become regarded as an attractive target for the screening of anti-influenza drugs. PA N-terminal (PA_N_) inhibitor can inhibit the entire PA endonuclease activity. In this study, we screened the effectivity of PA_N_ inhibitors from the FDA database through in silico methods and in vitro experiments. PA_N_ and mutant PA_N_-I38T were chosen as virtual screening targets for overcoming drug resistance. Gel-based PA endonuclease analysis determined that the drug lifitegrast can effectively inhibit PA_N_ and PA_N_-I38T, when the IC_50_ is 32.82 ± 1.34 μM and 26.81 ± 1.2 μM, respectively. Molecular docking calculation showed that lifitegrast interacted with the residues around PA or PA-I38 T’s active site, occupying the catalytic site pocket. Both PA_N_/PA_N_-I38T and lifitegrast can acquire good equilibrium in 100 ns molecular dynamic simulation. Because of these properties, lifitegrast, which can effectively inhibit PA endonuclease activity, was screened through in silico and in vitro research. This new research will be of significance in developing more effective and selective drugs for anti-influenza therapy.

## 1. Introduction

Influenza is an infectious disease that causes 300,000 to 500,000 human deaths globally per year [1,2]. The Influenza virus genome is composed of eight negative RNA segments encoded by multiple viral proteins, including PB2, PB1, PA, HA, NP, NA, M1, and NS1 [3,4]. RNA-dependent RNA polymerase (RdRp) is critical for virus RNA transcription and replication and comprises PA, PB1, and PB2 (Figure 1). The N-terminal of PA domain (PA_N_) contains an endonuclease active pocket and plays a crucial role in influenza virus polymerase activity. Drugs that abolish PA_N_ endonuclease activity or disturb the assembly of RdRp can effectively inhibit the replication of the influenza virus [5]. The PA_N_ endonuclease domain is highly conserved among different influenza virus subtypes, indicating that PA_N_ is a promising broad-spectrum anti-influenza therapeutic target because of its ability to inhibit virus proliferation during the initial mRNA synthesis stage [6].

The three classes of FDA-approved anti-influenza drugs are: neuraminidase inhibitor (oseltamivir and zanamivir), M2-ion channel inhibitor (adamantanes), and PA_N_ endonuclease inhibitor (Baloxavir acid). Several classes of PA_N_ endonuclease inhibitors researched included: 4-substituted 2,4-dioxobutanoic acids and 3,4-substituted 2,6-diketopiperazines [7], flutamide and derivatives [8,9,10], N-hydroxamic acid-scaffold compounds [11], catechins [12], etc. Baloxavir acid, the inhibitor targeted on the PA_N_ endonuclease, was approved by the FDA as an inhibitor of influenza A and B after proving effective in clinical trials [13].

A mutation in the influenza viral genome led to the influenza’s subsequent drug resistance. The hydrophobic interaction residues between PA_N_ endonuclease and Baloxavir acid were disturbed after the Ile 38 mutated to Thr [14]. The I38T substitution in the PA_N_ endonuclease domain is the primary mutant that leads to influenza’s resistance to Baloxavir acid. This resistance reduces the effectiveness of Baloxavir acid as an anti-influenza drug [15]. Using the I38T mutant as the drug screening target may help to develop and refine the next-generation endonuclease inhibitors.

Drug repurposing research can effectively identify new drug repurposing opportunities, quickly expand the drug market, and reveal new commercially valuable uses for existing drugs [16]. Some examples of drugs that have been successfully repurposed include thalidomide [17], sildenafil [18], bupropion [19], and fluoxetine [20]. These drugs are currently used for applications beyond their initially approved therapeutic indications. The combination of in vitro and in silico methods will increasingly be used in the discovery of novel medicine [21].

This research screened the effectivity of PA_N_ endonuclease inhibitors from the FDA-approved database through in silico methods and in vitro experiments. PA_N_ inhibitors can inhibit the entire PA_N_ endonuclease activity. Since the purpose of this study is to identify drugs that overcome resistance, we screened inhibitors that target both PA_N_ and mutant PA_N_ -I38T. Experimental tests have verified that the drugs lifitegrast and saquinavir have an inhibitory effect on PA_N_, while the drugs lifitegrast and conivaptan have an inhibitory effect on PA_N_-I38T. In addition, molecular docking shows the interaction mechanism between lifitegrast and the active site of PA_N_ or PA_N_-I38T. Research suggests that lifitegrast may be a potential anti-flu drug. Finally, the method employed in this work could be utilized as a fast and viable strategy for accelerating research in the treatment of influenza.

## 2. Results

### 2.1. Virtual Screening and Compounds Selection

To develop the new PA_N_ endonuclease inhibitor structure, we performed the virtual screening protocol based on the PA_N_ and mutant PA_N_-I38T structure. The residue Ile38 was not essential in either metal-ion binding or in catalytic activity. The I38 substitution does not block the endonuclease reaction but offers resistance to the inhibitor because the I38 side chain interacts with the Baloxavir acid. After the virtual screening, the compounds interacting with PA_N_ or PA_N_-I38T were sorted by affinity score. The top-ranked compounds listed in Appendix A were selected. Nine compounds closely interacted with PA_N_ and PA_N_-I38T as hit compounds. Finally, six compounds, saquinavir, conivaptan, lifitegrast, rifaximin, dutasteride, and lurasidone, were used for further basic gel endonuclease determination. In comparison, three compounds, Teniposide [22], Simeprevir [23], and Nilotinib [24], were filtered according to levels of cytotoxicity.

### 2.2. The Inhibitory Effect of PA_N_ or PA_N_-I38T with Compounds 

We identified the five compounds that could inhibit the endonuclease activity of PA_N_ or PA_N_-I38T, respectively. Next, we incubated different concentrations of compounds with PA_N_ or PA_N_-I38T and then detected the cleavage of a substrate (Figure 2 and Figure 3).

The results showed that the compound saquinavir could effectively inhibit the cleavage of PA_N_ substrate (Figure 2D), but not PA_N_-I38T substrate. Meanwhile, compound conivaptan can effectively inhibit the cleavage of PA_N_-I38T substrate (Figure 3D), but not PA_N_ substrate. However, compounds conivaptan, dutasteride, rifaximin, and lurasidone showed no inhibition of the endonuclease activity of PA_N_ (Appendix A); compounds dutasteride, rifaximin, saquinavir, and lurasidone showed no inhibition of the endonuclease activity of PA_N_-I38T (Appendix A). Finally, the compound lifitegrast was shown to effectively inhibit the substrate cleavage of both PA_N_ and PA_N_-I38T at 32.82 ± 1.34 μM and 26.81 ± 1.2 μM, respectively (Table 1).

### 2.3. Retained Stability of Conformations of PA_N_/PA_N_-I38T and Lifitegrast during MD Simulations

To gain further insight into the binding mode of the system after reaching equilibrium, 100 ns MD simulations were performed using an explicit solvent. The initial confirmation of lifitegrast was obtained from the optimal pose of molecular docking operations in virtual screening. The RMSD values are an essential parameter in assessing the stability of a protein–ligand complex. As shown in Figure 4A, the structure of both proteins and ligands acquired good equilibrium during 100.0 ns. Accordingly, we can draw the same conclusion from the radius of the gyration of protein analysis (Figure 4B).

Furthermore, to estimate the structural flexibility, the mean RMSF values were calculated. Figure 4C indicates that residues that interact with lifitegrast at the active pocket of protein remain stable. Although the residue Asn55-Leu71 in the Loop region is highly flexible, the overall protein conformation is stable (Figure 4D).

### 2.4. The Interaction Mechanism between PA_N_/PA_N_-I38T and Lifitegrast

Next, the protein–ligand complex structures in the 100 ns molecular simulation trajectories with similar conformations were divided into the same clusters. The representative frame of the largest cluster was extracted for analysis. Figure 5A,B show that lifitegrast binds to PA_N_/PA_N_-I38T at the structure active site and forms hydrogen bond interactions with lifitegrast (Figure 5C,D). Table 2 gives the detailed non-bond parameters of lifitegrast and PA_N_/PA_N_-I38T. Comparing the number of hydrogen bonds between protein and lifitegrast, it can be concluded that lifitegrast has three hydrogen-bond interactions with residues Arg124 of PA_N_, and four hydrogen-bond interactions with residues Trp88, Thr123, and Arg125 of PA_N_-I38T.

## 3. Discussion

Researchers have persistently searched for a drug that could be used to treat all types of influenza viruses, including influenza A, influenza B, and influenza C, along with their mutants. In this study, we used PA_N_ as the influenza drug virtual screening target because the PA endonuclease domain is highly conserved among different influenza virus subtypes. The I38T substitution in the PA_N_ endonuclease domain is the primary mutant that leads to resistance to Baloxavir acid (PA_N_ endonuclease inhibitor). Resistance develops when the hydrophobic interaction residues between PA_N_ endonuclease and Baloxavir acid are disturbed after the Ile 38 mutates to Thr. Therefore, we used the mutant type of PA_N_-I38T as the virtual screening target to avoid drug resistance. We used the FDA database to effectively identify drug repurposing opportunities. Then, we discovered a potent inhibitor, lifitegrast, which differs from Baloxavir acid in that it has the ability to interact equally with both PA_N_ and PA_N_-I38T.

Gel-based endonuclease inhibitory assay showed that lifitegrast effectively inhibits the substrate cleavage of both PA_N_ and PA_N_-I38T. Molecular simulation displayed that lifitegrast interacted with the active site of PA_N_ or PA_N_-I38T. Further molecular dynamic simulation analysis suggested that the bindings between PA_N_/PA_N_-I38T and lifitegrast were stable.

However, there are still numerous challenges. Subsequent viral experiments in vivo need to be implemented in order to verify the reliability of experiments in vitro. Meanwhile, lifitegrast has been used for the treatment of dry eye, but further toxic experimentation is required before lifitegrast can be used as a treatment for the influenza virus. 

In this research, we screened the effectivity of the PA_N_ endonuclease inhibitor from the FDA database through in silico and in vitro experiments. The findings suggest that lifitegrast may be a potential anti-flu treatment drug because it could solve the drug-resistant properties, which are superior to currently available flu treatments, such as Baloxavir acid. Finally, the method adopted in this work could be utilized as a fast and viable strategy for accelerating research on the treatment of influenza.

## 4. Materials and Methods

### 4.1. Virtual Screening

The PA_N_ endonuclease protein crystal structure (PDB ID: 6FS6) and I38T mutant (PDB ID: 6FS7) in the PA_N_ subunit were obtained from the RCSB Protein Data Bank database [25]. One thousand four hundred and ten compounds in mol2 format were downloaded from the FDA-approved drug database (updated in February 2018) for screening. The Raccoon program was used to convert the ligand from the mol2 format to the PDBQT format, which can be recognized by the virtual screening software [26]. Virtual screening was performed by the AutoDock Vina program [27], with the parameter values of x, y, z center set to 18.26, 95.54, 44.58, and the parameter values of the grid map in x, y, z-dimension set to 25 Å × 25 Å × 25 Å. Results of virtual screening were listed according to the binding energy score as shown in the Appendix A.

### 4.2. Chemistry

Compounds saquinavir, conivaptan, lifitegrast, rifaximin, dutasteride, and lurasidone were purchased from Aladdin (Shanghai, China) or Yuanye Biotech (Shanghai, China). Substrate DNA was purchased from Takara Bio lic. (Beijin, China)

### 4.3. Expression and Purified of PA_N_ and PA_N_-I38T

The PA_N_ endonuclease domain from influenza A type virus (H1N1) (PA_N_, residues 1-197) was cloned into the pET-28a vector, expressed in *E. coli* BL21 (DE3), and purified by IMAC. The positive clone was used for large-scale expression. Cells were expressed in LB medium at 37 °C for 4 h with 1 mM IPTG. Before the endonuclease activity assay, the protein sample was concentrated, and the buffer changed (10 mM Tris-HCl, 100 mM NaCl, and 2 mM MnCl_2_) by ultrafiltration. The protein concentration was measured by the BCA method. The PA_N_-I38T was cloned from the PA_N_ through site-directed mutagenesis. The mutant PA_N_-I38T was expressed and purified the same as the wild PA_N_.

### 4.4. Gel-Based Endonuclease Inhibitory Assay

The PA_N_ endonuclease activity was detected through the digestion of a single-strand DNA substrate. In the digestion reaction, PA_N_ or PA_N_-I38T was mixed with ssDNA in a digestion buffer that included 2 mM MnCl_2_. To determine the concentrate of PA_N_ or PA_N_-I38T for endonuclease activity assay, 0–1.5 μM protein and 100 ng ssDNA were incubated at 37 °C for 60 min. After digestion, the samples were fractioned by 0.8% agarose gel and visible by a stain. Image J was used to determine the density of substrate ssDNA. Finally, 1.5 μM PA_N_ or PA_N_-I38T were digested at 100 ng ssDNA in 10 μL volume for 60 min.

Compounds screened from the FDA library analysis determined the inhibition of endonuclease activity. In total, 1.5 μM PA_N_ or PA_N_-I38T and 100 ng ssDNA mixed with different concentrations of compounds were used for the experiment test. Samples without inhibitors were used as the no-compound control. Samples without protein were used as the substrate control. After digestion, the samples were loaded on the agarose gel for separation and were visible by a stain. The amount of remaining ssDNA was determined by image J. The IC_50_ value was calculated by GraphPad Prism 6.0 through non-Linear regression.

### 4.5. Molecular Dynamic (MD) Simulation

The MD simulation of PA_N_ and PA_N_-I38T with or without compounds was performed using GROMACS version 2018 with the CHARMM36 all-atom force field (28 March 2019) to investigate the inhibition mechanism [28]. The initial confirmation of lifitegrast was obtained from the optimal pose of the virtual screening results. The CGenFF server was used for the topology generation of the compound [29,30]. The protein was centered in separate cubic boxes and solvated using the SPC216 water model [31]. Two additional NA ions were added to the system to automatically achieve electron neutrality because the protein had a total charge of −2.000 e. The structure was relaxed through energy minimization to ensure a reasonable starting structure in terms of geometry and solvent orientation. Convergence was achieved at a maximum force of less than 1000 kJ/mol·nm in any atom. Equilibration was conducted in two phases, i.e., the NVT (constant number of particles, volume, and temperature) and the NPT (constant number of particles, pressure, and temperature) ensembles, for 100 ps until the system was equilibrated. A total of 100 ns simulations were performed after the equilibration. The root mean square deviation (RMSD), root mean square fluctuations (RMSF), and gyration radius were calculated using the tools available in the GROMACS package. According to the GROMOS method [32], RMSD is used as a metric, and the cluster analysis of molecular simulation trajectories is divided into different groups by the GROMACS package. For each system, the cutoff value of RMSD was set to 2.5 Å. The protein trajectory was displayed, animated, and analyzed through the PyMOL visualization program [33,34,35].

### 4.6. Statistical Assay

The IC_50_ was calculated by GraphPad Prism 6.0 using non-linear regression. The data were represented as mean ± SD.

## Figures and Tables

**Figure 1 molecules-26-07326-f001:**
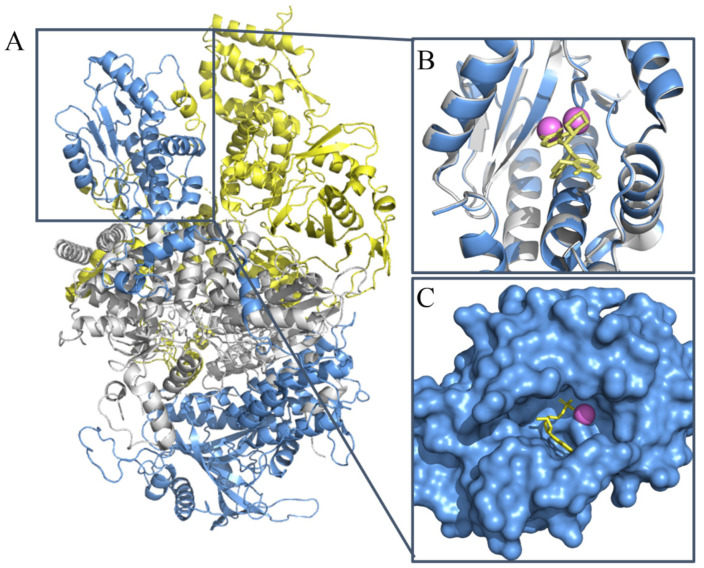
The structure of PA endonuclease. (**A**) The structure of RNA-dependent RNA polymerase (influenza A virus H5N1, PDB ID: 6QPF). The PA domain is shown in blue. The PB1 domain is shown in white. The PB2 domain is shown in yellow. (**B**) The cartoon structure of PA N-terminal (blue) (PDB ID: 6FS6) or PA-I38T N-terminal (white) (PDB ID: 6FS7) endonuclease domain complex with the inhibitor BXM. The BXM is shown in yellow. The Mn2+ is indicated with violet spheres. (**C**) The PA active site pocket complex with the substrate Amp. Amp is shown in yellow. The Mn2+ is indicated with violet spheres.

**Figure 2 molecules-26-07326-f002:**
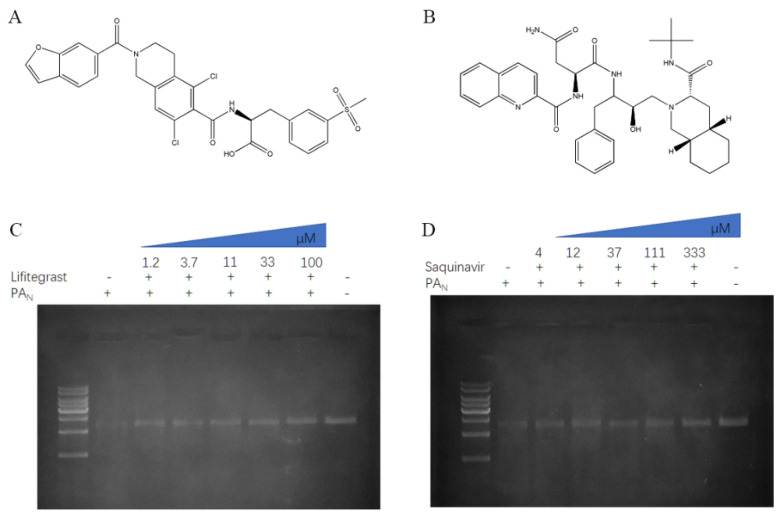
Compound inhibition of the endonuclease activity of PA_N_. The chemical structure of lifitegrast (**A**) and saquinavir (**B**). For the inhibition assay, different concentrations of the compounds lifitegrast (**C**) and saquinavir (**D**) were incubated with the 1.5 μM PA_N_ and 100 ng ssDNA at 37 °C for 1 h. After the digestion, the products were resolved on agarose gel.

**Figure 3 molecules-26-07326-f003:**
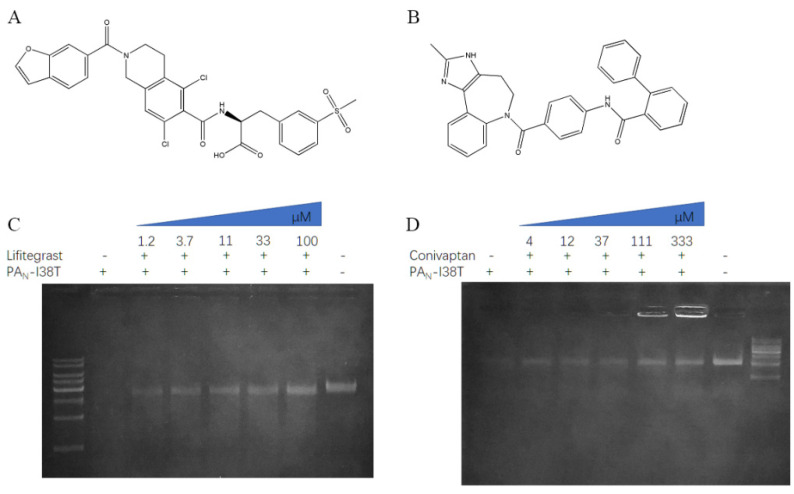
Inhibition of the endonuclease activity of PA_N_-I38T. The chemical structure of lifitegrast (**A**) and conivaptan (**B**). For the inhibition assay, different concentrations of compounds lifitegrast (**C**) and conivaptan (**D**) were incubated with the 1.5 μM PA_N_-I38T and 100 ng M13mp18 at 37 °C for 1 h. After the digestion, the products were resolved on agarose gel.

**Figure 4 molecules-26-07326-f004:**
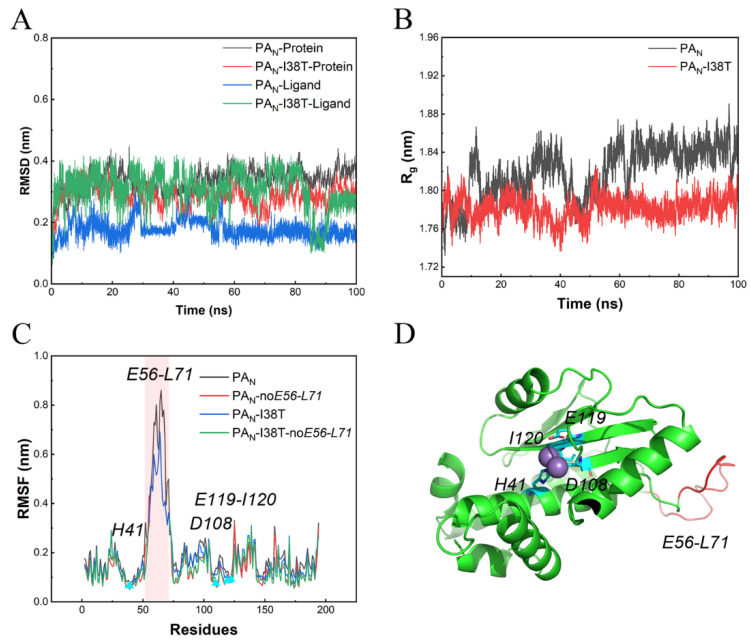
RMSD (**A**), Rg (**B**) and RMSF (**C**) propensities of PA_N_ and PA_N_-I38T with ligand during molecular dynamic simulation. The highly flexible residue Asn55–Leu71 in the Loop region is colored red. The residue His41, Asp108, Glu119, and Ile120 associated with the active site is colored blue (**D**) and is stable.

**Figure 5 molecules-26-07326-f005:**
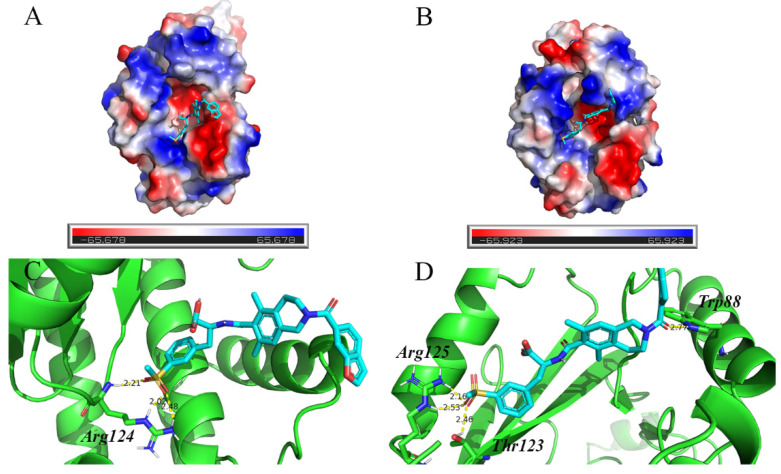
The interaction of lifitegrast within PA_N_ and PA_N_-I38T active sites. The representative structure is extracted from the largest number of clusters in the system after molecular dynamics simulation. Electrostatic potential surface of PA_N_ (**A**) or PA_N_-I38T (**B**) structure with lifitegrast in the active site pocket. Structure of PA_N_ (**C**) or PA_N_-I38T (**D**) with lifitegrast. Manganese ions are indicated as gray spheres. Lifitegrast is shown using blue sticks. PA_N_ or PA_N_-I38T are shown as green cartoons. Hydrogen bonds are shown as yellow dashed lines.

**Table 1 molecules-26-07326-t001:** IC_50_ values of compound inhibited PA_N_ and PA_N_-I38T endonuclease.

Chemical Name	PA_N_ IC_50_ (μg/mL) ^1^	PA_N_-I38T IC_50_ (μg/mL)
lifitegrast	32.82 ± 1.34	26.81 ± 1.2
conivaptan	NI	227.7 ± 1.33
saquinavir	372.7 ± 1.38	NI

^1^ Measurement IC_50_ values of compounds through five independent experiments and data were shown as mean ± SD.

**Table 2 molecules-26-07326-t002:** Hydrogen bond parameters of lifitegrast and PA_N_/PA_N_-I38T.

PA_N_			PA_N_-I38T		
Donors Atom	Receptor Atom	Distances (Å) ^1^	Donors Atom	Receptor Atom	Distances (Å)
Arg124:NH	lifitegrast:O4	2.21	Trp88:HE1	lifitegrast:O24	2.77
Arg124:HE	lifitegrast:O3	2.02	Thr123:HG1	lifitegrast:O3	2.46
Arg124:1HH2	lifitegrast:O3	2.48	Arg125:HE	lifitegrast:O3	2.16
			Arg125:1HH2	lifitegrast:O3	2.56

^1^ The length of the hydrogen bonds.

## Data Availability

Not applicable.

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
