# Peer review of "Drug Repurposing for Influenza Virus Polymerase Acidic (PA) Endonuclease Inhibitor"

_molecules, 2021, doi:10.3390/molecules26237326_

Round 1
Reviewer 1 Report
Authors identified the repurposing drug lifitegrast as the inhibitors of PA endonuclease of influenza A viruses using molecular docking and in vitro assays of recombinant PAN and mutant PAN-I38T (drug resistant variant). They showed the iC50 value of lifitegrast 32.82±1.34 μM for wild type PAN and 26.81±1.2 μM for PAN-I38T, respectively. The approach will useful to develop effective and selective drugs.
- Authors should the quantitative data for Figure 2 and Table 1. How to quantitate the PAN activity?
- Authors should provide the data about the activity of PAN mutants with Lys137Ala and its interaction with lifitegrast.
Reviewer 2 Report
November 13, 2021
Manuscript Number: molecules-1468825
Title: Drug repurposing for influenza virus polymerase acidic (PA) endonuclease inhibitor
Authors: Xin Meng , Ye Wang
Overview and general recommendation:
The authors conducted the screening of effective PAN inhibitors from the FDA database using in silico approach, including both docking and molecular dynamics simulations, followed by in vitro experiments. They propose Lifigerast as an effective inhibitor of PA endonuclease activity in anti-influenza therapy. Given the issue of drug resistance in anti-influenza treatment, the development of inhibitors that exhibit broad-spectrum anti-influenza virus activity is important and has received broad interest. The results are well supported by the data, the data are well presented, and the manuscript is well written. The paper is acceptable for publication after minor revision.
Major comments:
- The presented RMSDs of ligands show some variations, in particular for the I38T mutant. I wonder if the bound poses in the simulations are different from those shown in Figure 4. It is recommended that the authors analyze the bound poses in the simulations, such as by clustering analysis, and provide the simulation-derived poses as well. Those poses may have better quality, at least in principle, than those seen in Figure 4 that were obtained from the docking simulations.
- The authors can also calculate the RMSF without the E56-L71 region to better quantify the difference in the fluctuation around the binding site. RMSF with irrelevant motions sometimes obscure the proper interpretation of the results.
- The authors can benefit from expanding the discussion section. The value of the presented results and methodology can be discussed under the comparison with precedent works. For example, it seems that the presented candidate, Lifigerast, is advantageous over the existing one, such as Baloxavir acid, in terms of drug resistance. The authors can discuss the point in more details.
Minor comments:
- The statement of “Lower values of RMSD indicate that the equilibrated system is more stable.” may be misleading. RMSD is a measure of the deviation from the reference structure. When RMSD is calculated with respect to an initial structure, the lower the value, the better the initial model. Instead, RMSD is often used to measure whether the equilibrium have been reached (RMSD converges to around some value in equilibrium).
- In Figure 5B, the unit of time can be changed from “ps” to “ns”.

Round 2
Reviewer 1 Report
In the revision, authors reply the comments about the molecular docking, but did not reply my comments about the quantitative data, including
- Authors should the quantitative data for Figure 2 and Table 1. How to quantitate the PAN activity?
- Authors should provide the data about the activity of PAN mutants with Lys137Ala and its interaction with lifitegrast.
Author Response
Please see the attachment

This manuscript is a resubmission of an earlier submission. The following is a list of the peer review reports and author responses from that submission.